# A STEDable BF_2_-Azadipyrromethene Fluorophore for Nuclear Membrane and Associated Endoplasmic Reticulum Imaging

**DOI:** 10.3390/membranes15010009

**Published:** 2025-01-01

**Authors:** Anaïs C. Bourgès, Massimiliano Garre, Dan Wu, Donal F. O’Shea

**Affiliations:** Department of Chemistry, RCSI, University of Medicine and Health Sciences, 123 St Stephen’s Green, D02 YN77 Dublin, Ireland

**Keywords:** nuclear membrane, endoplasmic reticulum, STED microscopy, fluorophores, BF_2_-azadipyrromethene, nuclear pores, nuclear lamina, super resolution imaging, DNA

## Abstract

The endoplasmic reticulum and the internal nuclear compartments are intrinsically connected through the nuclear membrane, pores and lamina. High resolution imaging of each of these cellular features concurrently remains a significant challenge. To that end we have developed a new molecular nuclear membrane-endoplasmic reticulum (NM-ER) staining fluorophore with emission maxima at 650 nm. NM-ER is compatible with fixed and live cell imaging and stimulated emission depletion microscopy (STED) showing significant improvement in resolution when compared to comparable confocal laser scanning microscopy. The imaging versatility of NM-ER was illustrated through its compatible use with other fluorophores for co-imaging with DNA, nuclear pores and lamina allowing cellular abnormalities to be identified. NM-ER alone, or in use with other nuclear region labels could be an important tool for the investigation of nuclear transport and associated cellular processes.

## 1. Introduction

The nuclear envelope (NE) separates the contents of the nucleus from the cytoplasm and provides the structural framework of the nucleus. It acts as the barrier to prevent free passage of molecules between the nucleus and the cytoplasm, thereby maintaining the nucleus as a distinct biochemical compartment. The nucleus is surrounded by a system of two concentric membranes, called the inner and outer nuclear membranes (NM). The outer nuclear membrane is continuous with the endoplasmic reticulum (ER) [1], while underlying the inner membrane is the nuclear lamina, a fibrous meshwork that provides structural support to the nucleus and can act as sites of chromatin attachment [2,3,4,5,6]. The sole channels through the nuclear membrane are provided by the nuclear pore (NP) complexes, which allow the regulated exchange of molecules between the nucleus and cytoplasm [7]. The selective traffic of proteins and RNAs through the NPs not only establishes the internal composition of the nucleus but also plays a critical role in regulating gene expression.

Fluorescence microscopy is ideally suited to investigate the nucleus and its closest interconnected environments and the dynamic events of cell replication [8]. Also, identification of deviations from normal nuclear morphologies are important indicators of diseases such as cancer or premature aging syndrome [9]. Advances in super resolution fluorescence imaging continue to deliver a greater understanding of cellular structures and processes with an extensive volume of ongoing research [10]. Stimulated emission depletion (STED) nanoscopy represents one such super-resolution method in which the challenge of diffraction limited resolution is overcome [11]. In STED imaging, an excitation laser (EL) is employed in conjunction with a second doughnut-shaped depletion laser (DL) which de-excites fluorophores in the partially coinciding laser regions through a stimulated emission at the wavelength of the DL. This provides a higher resolution image from the residual area of non-overlapped lasers [12]. The far-red region of the spectrum is useful for biological fluorescence imaging since there is a greater penetration of light at longer wavelengths, a lower phototoxicity, and cellular autofluorescence is minimal [13]. With the development of the 775 nm near-infrared DL, longer wavelengths for STED imaging are now possible, enabling improved resolution imaging of live cells [14].

While STED imaging of nuclear lamina and the ER have recently been reported with labelled antibodies for immunofluorescence or with green fluorescent protein [15,16], and several dyes have been developed for NPs [17] and DNA imaging [18], to date no STED compatible molecular fluorophore has been developed for imaging the lipid NM. Both the inner and outer nuclear membranes are phospholipid bilayers, which, like other cell membranes, are permeable to small nonpolar molecules so could be amenable to imaging with a suitable molecular fluorophore. It would be of significant advantage if such a fluorophore was live and fixed cell STED compatible and could be utilized in conjunction with co-imaging of other interconnected features such as DNA, NPs and lamina.

In this report, we describe a new far-red emitting molecular fluorophore 1 as a NM and ER marker and assess its super-resolution imaging performance in conjunction with co-labelling nearby cellular features (DNA, NPs, and lamina) with available commercial fluorophores (Figure 1). The co-imagers utilized being Hoechst 33342 or NucSpot650 for nuclear DNA, wheat germ agglutinin (WGA) conjugate Alexa 647 or CF594 for NPs, and lamin B1 antibody conjugated to CF594 for the lamina.

## 2. Materials and Methods

Synthesis of 3-(nitromethyl)-1-phenyl-6-((tetrahydro-2H-pyran-2-yl)oxy)hexan-1-one **4**. A solution of compound **3** (2.64 g, 9.63 mmol) nitromethane (5.2 mL 96.3 mmol) and diethylamine (DEA) (4 mL, 38.6 mmol) in EtOH (50 mL) was heated under reflux for 16 h. The reaction mixture was cooled to rt, water added (50 mL) and extracted with EtOAc (2 × 50 mL), washed with water (25 mL), brine (25 mL) and dried over Na_2_SO_4_. Solvent was evaporated and the residue purified by silica gel chromatography (eluent: petroleum ether/EtOAc, 70:30) to give **4** (2.6 g, 82%) as an oil. ^1^H NMR (400 MHz, CDCl_3_): δ 7.95 (m, 2H), 7.59 (m, 1H), 7.48 (m, 2H), 4.63–4.51 (m, 3H), 3.88–3.71 (m, 2H), 3.53–3.36 (m, 2H), 3.24–3.16 (m, 1H), 3.12–3.05 (m, 1H), 2.96–2.86 (m, 1H), 1.85–1.75 (m, 1H), 1.73–1.65 (m, 3H), 1.63–1.50 (m, 6H) ppm. MS (ESI): *m*/*z* [M + H]^+^ calcd for C_18_H_26_NO_5_ 336.4, found 336.5. 

Synthesis of 3,3′-(5,5-difluoro-3,7-diphenyl-5H-4l4,5l4-dipyrrolo [1,2-c:2′,1′-f][1–3,5]triazaborinine-1,9-diyl)bis(propan-1-ol) **1**. Compound **4** (745 mg, 2.22 mmol) and NH_4_OAc (6.0 g, 77.6 mmol) in MeOH (40 mL) was heated under reflux for 9 h. The reaction mixture was allowed to cool to rt; water (100 mL) added and extracted with CH_2_Cl_2_ (75 mL × 2). Combined organic layers were washed with saturated NaHCO_3_ (25 mL), water (25 mL), brine (25 mL) and dried over Na_2_SO_4_. Solvent was evaporated and the residue passed through a silica column (eluant: *c*-C_6_H_12_/EtOAc, 90:10) to give a purple solid of **5** (270 mg, 0.46 mmol, 36%). Without further purification, **5** (50 mg, 0.086 mmol) was dissolved in CH_2_Cl_2_ (10 mL), treated with diisopropylethylamine (150 µL, 0.86 mmol) and BF_3_.Et_2_O, (212 µL, 1.72 mmol) and stirred under N_2_ for 90 min. Solvent was evaporated and the residue purified by silica gel chromatography (eluant: DCM/MeOH, 90:10) to give NM-ER **1** (24 mg, 0.052 mmol, 60.4%) as a green metallic solid. ^1^H NMR (400 MHz, CDCl_3_): δ 8.01–7.94 (m, 4H), 7.46 (m, 6H), 6.63 (s, 2H), 3.70 (t, J = 5.8 Hz, 4H), 2.91 (t, J = 7.2 Hz 4H), 1.98–1.88 (m, 4H) ppm. ^19^F NMR (376 MHz, CDCl_3_): δ −132.36 ppm. MS (ESI): *m*/*z* [M + Na]^+^ calcd for C_26_H_26_BF_2_N_3_O_2_Na 484.3, found 484.4. 

Microscope set-up. Microscopy CLSM and FLIM images were acquired on a Leica Stellaris 8 Falcon microscope (Wetzlar, Germany) fitted with NKT Photonics White Light Laser (440–790 nm) and controlled by LAS X (version. 4.4.0.24861) (as described previously in [19]). Live cell imaging experiments were carried out under Okolab incubation system to maintain the temperature at 37 °C and CO_2_ at 5%. Images were acquired using a Leica HC PL APO CS2 100X/1.40 oil immersion objective. Excitation wavelength was obtained tuning the White Light Laser (WLL) with a maximum power used for cell imaging of 0.67 µW at 594 nm (~0.3 µW usually). For STED imaging, the depletion laser at 775 nm was used at 20% (0.21 mW) except for NucSpot650 at 10% [20]. A 405 nm laser was used for the excitation of Hoechst at a maximum power of 0.3 µW. Images were acquired at 200/400 Hz with an exposure time of ~1.5–2 µs / pixel in confocal and ~3–5 µs/pixel in STED. A HyD (S2 or X3) detector collected emission of all the red emitting dyes in counting mode and a HyD S1 detector in analog mode collected emission from Hoechst (Table 1). TauSTED mode from Leica Microsystem (Tau Strength 100, Denoise 50 and time gate 0.5–11.5 ns) was chosen to acquire the STED images of all the dyes except Alexa647. Line accumulation of usually 4 lines was used for STED images. 3D imaging was done with z steps of 0.3 µm in confocal and 0.18 µm in STED. The FLIM function was used to acquire the real time fluorescence lifetime imaging in solution or in the cells. Emission spectra were acquired by collecting the emission with a step size of 3 nm and a bandwidth of 5 nm.

Data processing. All images were processed using Leica LasX software (version 4.6.0.27096) with Lightning post processing applied with three iterations. ImageJ 1.54f was used to calculate the FWHM (σ) from a gaussian fit, the ImageJ plugin JaCoP for colocalization [21] and Igor Pro 8.04 to plot the graphs. LASX alcon (version. 4.4.0.24861)was used for lifetime and phasor analysis. FLIM and phasor plot analysis were performed on the same data set. We define as FLIM and the FLIM image the time-domain technique that uses time-correlated single photon counting (TCSPS) to measure fluorescence lifetime from fitting the fluorescence lifetime decay. The FLIM image displays the average fitted lifetime of all the photons collected for each pixel (color coded accordingly). On the other hand, phasor lifetime analysis is model-free technique that relies on Fourier transformation (frequency-domain technique) where a pixel of an image becomes a dot on a 2D phasor plot [22,23,24]. A cluster of dots will have a similar lifetime which is how the phase lifetimes were determined. For co-stained samples, the cloud of dots is usually very stretched with pixels/dots being a mix of the 2 fluorophores. To separate them, circles can be positioned manually where the 2 “pure” lifetimes should be, usually at both edges of the cloud of dots, and then 2 channels are artificially created (with an overlay).

Cell Culture. MDA-MB 231 and HeLa Kyoto cells were cultured in Dulbecco’s Modified Eagles Medium supplemented (DMEM) with 10% fetal bovine serum (FBS), 1% L Glutamine, and Penicillin/Streptomycin (1000 U/mL) (DMEM+++), and incubated at 37 °C and 5% CO_2_. Cells were seeded on to an eight well chamber slide (Ibidi) at a density of 1 × 10^4^ cells per well for 48 to 72 h (about 80% confluency) as described in [19].

Live cells experiment. Cells were washed one time with 250 μL of prewarmed DMEM+++. Cells were stained with 0.4 to 1 μM of NM-ER **1** for at least 30 min. Nuclear DNA was stained with 3 μM of Hoechst 33342 (62249, Thermo Scientific, Waltham, MA, USA) for at least 30 min. For NucSpot 650/665, media was replaced by 150 μL of a 1/1000 dilution of the dye in prewarmed DMEM+++ as recommended by the supplier (41034T, Biotium, Fremont, CA, USA) for at least 40 min. For co-staining NucSpot650 was stained first. For ER Green Tracker (E34251, Invitrogen, Waltham, MA, USA), media was replaced by a 1 µM solution in prewarmed DMEM+++ for an hour and washed. Cells were imaged directly.

Fixed cells experiment. Cells were washed three times with prewarmed PBS and fixed for 20 min with a prewarmed 4% PFA solution (in PBS). Cells were washed and could be stored at 4 °C until needed. Staining was performed with 0.5–1 μM of NM-ER **1**, ≈5 μg μL of WGA-CF594 (Wheat Germ Agglutinin CF594 conjugate, 29023-1, Biotium)/WGA-Alexa647 (W32466, Invitrogen), 3 μM of Hoechst for at least 30 min at RT or NucSpot650 (1/1000 dilution in PBS). Cells were imaged directly.

Cell permeabilization and antibodies stain. Fixed cells were permeabilized for 6 min with 0.2% of TritonX (in PBS) at RT and washed three times. A blocking step was done for at least 1 h with a 3% BSA solution (in PBS) at RT. Cells were then incubated for 90 min at RT with Lamin B1 polyclonal antibody (rabbit IgG, PA5-19468, Invitrogen) diluted 1/1000 in PBS with 3% BSA as described in [15]. After washing, the cells were incubated for 1 h at RT with the secondary antibody anti-rabbit IgG CF594 produced in goat (SAB4600107, Sigma-Aldrich, St. Louis, MO, USA) diluted 1/1000 in PBS with 3% BSA. Cells were washed in PBS and imaged.

In solution measurement. The different fluorophores were diluted to a concentration of 25 μM in PS20-H_2_0 for NM-ER **1** and NIR-AZA **2** and PBS for the others. Measurements were performed in an eight well chamber slide in a total volume of 250 μL at RT. See Table 2 for the emission and detection wavelengths used for each fluorophore as well as their lifetime measured at 20% of WL.

## 3. Results and Discussion

### 3.1. Design, Synthesis and Photophysical Characterization of NM-ER ***1***

The BF_2_-azadipyrromethenes are a highly adaptable category of fluorophore with uses in the material, chemical, biological and medical sciences [25]. Their photophysical wavelengths can be refined to reside within the red to the near-infrared spectral range of 600 to 1060 nm which allows their uses to span from solar cells to cell microscopy. Derivatives of these fluorophores have previously been used for a variety of in vitro and in vivo fluorescence imaging applications [26,27,28,29,30,31,32] but their first adaption to super resolution STED imaging has only recently been reported [33]. It was shown that a specific derivative, with absorbance and emission maxima at 623 and 648 nm respectively and fluorescence lifetime of 3.6 ns, was responsive to a 775 nm DL giving STED images with the expected resolution improvements over confocal images. Additionally, at the concentrations used for live cell imaging, it demonstrated excellent photostability with no cellular toxicity.

For this work, the key twofold criteria for the structural design of **1** was to ensure effective accumulation within the NE and ER and to have STED compatible photophysical characteristics. Inclusion of two primary alkyl alcohol groups was to provide balanced lipophilicity for effective localization of **1** in the NM and ER. The moderately lipophilic, non-charged, **1** would match the lipid rich microenvironment of the nuclear double membrane while disfavoring its accumulation in the plasma membrane. The role of the two phenyl rings being to tune the emission to lie in 600 to 750 nm range which would allow its modulation using a 775 nm STED depletion laser [33]. The synthesis of **1** relied on an adaptation of a previously reported route to this substitution pattern of BF_2_-azadipyrromethenes [34]. The tetrahydropyran protected α,β-unsaturated ketone **3** was subjected to diethylamine (DEA) basic conditions to affect a conjugate addition of the reagent nitromethane to provide **4** in a good 82% yield (Figure 2). Next compound **4** was heated in methanol under reflux in the presence of excess ammonium acetate to give the azadipyrromethene **5** which was converted to **1** through its reaction with boron trifluoride with diispropylethyl amine (DIEA) as base. Purification by silica gel chromatography competed the synthesis giving **1** as a metallic green solid with NMR and mass spectrometry analysis consistent with the expected structure.

As anticipated the emission of **1** spanned from 600 to 750 nm following an excitation at 595 nm and with good quantum yield (Figure 2). The fluorophore displayed only minor solvatochromic shifts with the absorption maxima in polar ethanol and lipophilic triolein being 615 and 630 nm with the corresponding emission maxima at 638 and 650 nm respectively.

### 3.2. Photophysical Characterizations for STED Imaging

Examination of the STED photophysical characteristics of NM-ER **1** was done by comparison with the other fluorophores used in this study namely, Alexa647 and CF594 (Figure 3). While the parameters for optimal STED performance can be elusive it was instructive to make such cross comparisons at the outset of the work. It has been previously shown that fluorophores with absorption and emission maxima close to the wavelength of the DL display anti-Stokes interferences and cross-excitation which can compromise image quality [35,36,37]. As such the NIR-AZA **2** (Ar = pMeOC_6_H_4_) (absorption 700 nm, emission 732 nm) was included in this initial testing as it has been shown to produce anti-Stokes emissions with a 775 nm depletion laser and would allow the selected imaging fluorophores to be evaluated against it (Figure 1) [33].

NM-ER **1** and NIR-AZA **2** were characterized in an aqueous-lipid solution (H_2_O-polysorbate 20) and compared to Alexa647 and CF594 in PBS. Emission spectra, measured on the microscope, confirmed maxima emissions between 615–665 for **1**, Alexa647 and CF594 whereas **2** was at 720 nm (Figure 3a). The lifetime in solution of each fluorophore was measured using phasor lifetime imaging. This approach converts the time-resolved decay of fluorophores into a phasor plot such that each pixel of an image is represented on the plot [23,24] (see data processing in Section 2. Materials and Methods). This model-free analysis allows an easy visualization of the lifetimes especially for complex samples with overlapping lifetimes, multi-exponential decays or when STED depletion is applied. The phase lifetimes ranged from 1.2 for Alexa-647 to 3 ns for **1** and as expected, they remained unchanged as the percentage excitation laser (EL) power varied between **1** and 20% while the fluorescence intensity increased linearly (Figure 3b). Next, the same measurements were performed with the 775 nm depletion laser (DL) set at 20% power resulting in significantly shorter lifetimes for all fluorophores except NIR-AZA **2** (Figure 3c). This indicated that an effective de-excitation through STED had occurred for all the fluorophores except for **2** due to its strong anti-Stokes emission. Indeed, only in the case of fluorophore **2** were increasing photon counts detected in response to varying the DL power alone, i.e., in the absence of the EL (Figure 3d). Fluorophore **2** again showed its marked difference from **1** and the commercial dyes as their response to increasing EL power at constant 20% DL power was near linear, whereas **2** gave a constant unchanging high photon count (due to its anti-Stokes emission) (Figure 3c). As our new NM-ER fluorophore **1** fulfilled the criteria for STED imaging in a manner consistent with established STED dyes, we next turned our attention to its use in cell imaging alone and in tandem with co-staining other cell features.

### 3.3. Characterization of NM-ER ***1*** in Live and Fixed Cells for Super Resolution STED Imaging

Fluorescence imaging potential of NM-ER **1** was characterized in MDA-MB 231 (MDA) and HeLa Kyoto (HeLa) cancer cell lines. The confocal images revealed that subcellular structures stained, with 0.5 µM of the fluorophore, were the nuclear membrane with invaginations visible inside the nucleus and the endoplasmic reticulum (Figure 4a,b). Its pattern highly colocalized with the commercially available ER-Tracker Green (Appendix A) [2]. The same features were visible in live or fixed cells regardless of whether staining was before or after fixation. When 20% DL power was applied for STED imaging, a clear shift of the pixels towards shorter lifetimes was visible on the phasor plot (Figure 4d). Comparison of the field of view (FOV) in confocal (EL only) and STED (EL + DL) showed a marked increase in resolution with, for instance, the NM measuring three-fold smaller thickness averaged at 85 (10) nm. This improvement in resolution also allowed invagination tube structures which transect the nuclear body to be now visualized in STED and not in confocal (ROI 1) (Figure 4e). Its robustness against photobleaching was demonstrated in live and fixed cells, and in comparison with ER-Tracker Green (Appendix A). Notably following the acquisition of fifty STED images of the same FOV only negligible photobleaching had occurred which has allowed us to perform 3D STED imaging of cells (Appendix A and Appendix A). This stability advantage allows long term imaging over eight hours while also demonstrating the low cytotoxic effect of the fluorophore **1** as cells continue to divide normally (Appendix A).

### 3.4. Identification of Cancer Cellular Nuclear Abnormalities

Recognition and quantification of nuclear abnormalities is of importance in biomedical research as nuclear morphology is a common marker for cell determination and classification [38]. Indeed, abnormalities in shape, enlargement, irregular contours or micronuclei are observed in cancer cells and used for diagnosis [38]. A common approach being to utilize a DNA stain alone or in combination with a nuclear lamina stain [10]. Lamina staining requires the use of antibodies preventing their use in live cells or in combination with genetically encoded fluorophores which requires additional steps (cell transfection). As such we chose to investigate NM-ER **1** as a plausible alternative to picturing known cancer cell abnormalities which would be suitable for both live and fixed cells with no antibodies or cell permeabilization needed. Therefore, cells were co-stained with NM-ER fluorophore and a DNA nuclear dye. Our first choice was Hoechst, a commonly used DNA marking dye suitable for both live and fixed cell experiments. This combination allowed the visualization of the nuclear abnormalities mentioned above, nuclear invaginations, abnormal nuclear envelope shape or the presence of micronuclei in live and fixed cells (Figure 5 and Appendix A). Of particular note were the micronuclei which were relatively abundant in MDA cells and as a hallmark of genetic instability are frequently associated with cancer diseases [39,40].

These features were also assessed in super resolution using STED, however, as Hoechst is not compatible for STED imaging (Appendix A), the commercial dye NucSpot650 was employed in co-staining experiments with **1** [41]. Having both the NM and nucleus independently stained with STEDable dyes allowed an unobstructive confirmation that a distinct separation between **1** and nuclear DNA existed (Figure 6a–c). As NucSpot 650 and **1** did not colocalized 3D renditions of the NM closely circumnavigating the DNA staining pattern with high fidelity could be obtained (Figure 6).

Importantly, this gain in resolution also allowed the observation and distinction of small invaginations and micronuclei. Line analysis of both revealed NM invaginations did not contain DNA whereas micronuclei were isolated sites of DNA surrounded by their own NMs (Figure 6c) [42].

### 3.5. Co-Imaging of NM-ER with Nuclear Pores and Lamina

NPs act as the gatekeeper for nuclear entry and exit, being open channels embedded within and transecting from the outer through to the inner NM [43]. Several fluorophore conjugates of WGA have been previously reported for STED imaging of NPs (e.g., WGA-Alexa647 and WGA-CF594) [44,45] and so it was of interest to gauge their use with **1**. The aim being to confirm their spatial positioning across the NM and with NucSpot650 to show that they are not colocalized with the DNA label. Fixed cells were treated with **1** and WGA-Alexa647 for 30–60 min and imaged directly. Initially technical difficulties were encountered due to the leaking through of NM-ER emission into the detection channel of Alexa647. This was resolved by changing the recommended excitation and detection of Alexa647 to the longer wavelengths of 680 nm and 690–750 nm respectively, allowing both confocal and STED co-images to be acquired. Encouragingly, the co-staining experiments showed that the emission from **1** throughout the NM colocalized with the spotted Alexa647 staining pattern of the NPs within the membrane with improved resolution achieved when using STED over confocal images (Figure 7a,b). Because of NucSpot650 and Alexa647 overlapping emissions, we also imaged the nuclear DNA and the NPs with a well performing STEDable dye, CF594 (WGA conjugated). NPs are, on average, measured slightly larger than the thickness of the measured NM at 90 nm which is in agreement with other reports [45] (Appendix A) and they are clearly surrounding the nuclear DNA, as shown by NucSpot650, but without any colocalization (Figure 8a–c).

In addition to spectral wavelength separation, the different phase lifetimes of **1** and WGA-Alexa647 (i.e., 3.1 ns for **1** in the NM and 1.2 ns for Alexa647 in the NPs) were used to gate the two emissions from each other. Using FLIM (fitting of the fluorescence lifetime decay) and the phasor analysis the two fluorophores were distinguishable from each other (Figure 7). The confocal and STED FLIM images show that WGA-Alexa647 has a much shorter lifetime in the plasma membrane and NM-ER has a longer lifetime in the ER. Within the nuclear envelope the contrast between the two is not as clear. Separating them using the phasor plot was only possible in confocal imaging.

Overall, these experiments indicated that the NPs stained by WGA and located within the nuclear membrane are co-localized with **1** in the NM. The observable NPs consistently lie across the region stained by **1**, thereby confirming that our new fluorophore indeed stained both the inner and outer nuclear membranes (Figure 7b,c). Additionally, as no breaks in the NM staining by **1** are noted, this implies that it is also located inside the NPs themselves yet does not accumulate within the nucleus.

Next the nuclear lamina which underlies the inner NM was investigated for which antibodies against Lamin B1 CF594 conjugate were utilised. A layer of lamins was visible around the nuclear DNA, though not as smooth and continuous as the NM stained by NM-ER. In STED this layer, was not colocalized with NucSpot650 and appeared thinner in some sections at ≈ 60 nm (others sections measured up to 110 nm) than the NM measured with **1** (Figure 8 and Appendix A). This compared favorably with previous reported values of ~120 nm and ~130 nm [15,46]. Unfortunately, co staining with NM-ER was not as successful as the NM permeabilization, which is essential for the antibody based lamin staining, affected the staining of **1** causing it to have a low emission in comparison with lamin-CF594 (Appendix A).

## 4. Conclusions

A new BF_2_-azadipyrromethene fluorophore, NM-ER, designed to promote accumulation in internal lipid membranes, with optimal photophysical characteristics for super resolution STED imaging has been developed. When tested for live and fixed cell imaging in two cancer cell lines, it showed excellent accumulation in the inner and outer nuclear membranes and the endoplasmic reticulum. Co-staining experiments showed NM-ER to be compatible with other known DNA (Hoechst, NucSpot650) and nuclear pore complex (Alexa647) markers allowing the 3D spatial positioning of each nuclear feature to be imaged. Our results revealed that cancer cell abnormalities such as nuclear invaginations and micronuclei could be readily detected, with high resolutions, without the need for anti-body lamina staining. Taken together, these results show that the NM-ER fluorophore has substantial potential as a marker of key membrane features associated with the nucleus, further investigations of its potential uses are currently ongoing and will be reported on in due course.

## Figures and Tables

**Figure 1 membranes-15-00009-f001:**
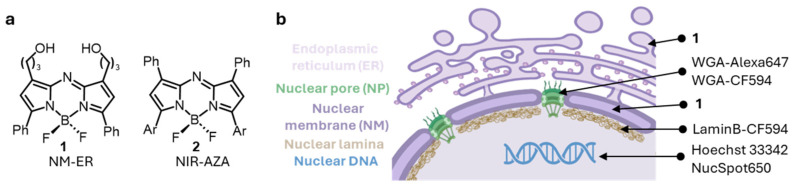
Overview of the fluorophores used and the imaging goals for this work. (**a**) Structures of the NM-ER fluorophore **1** and NIR-AZA **2**. (**b**) Schematic of the nucleus with the different structures stained and the list of dyes used for this work (drawn using BioRender).

**Figure 2 membranes-15-00009-f002:**
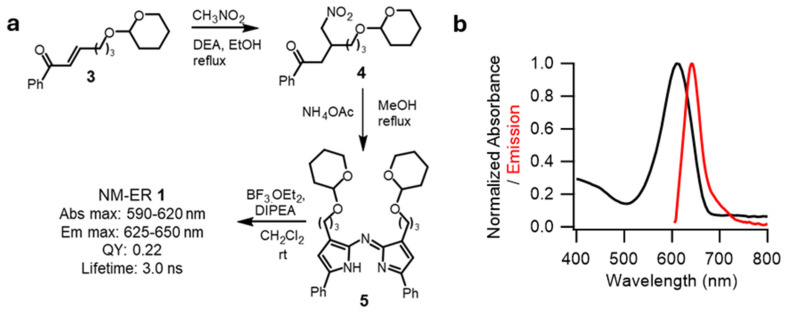
Development of STEDable NM-ER **1** fluorophore. (**a**) Synthetic route utilized for the making of **1**. (**b**) Absorption and emission spectra of NM-ER **1** in ethanol (5 μM, excitation 595 nm).

**Figure 3 membranes-15-00009-f003:**
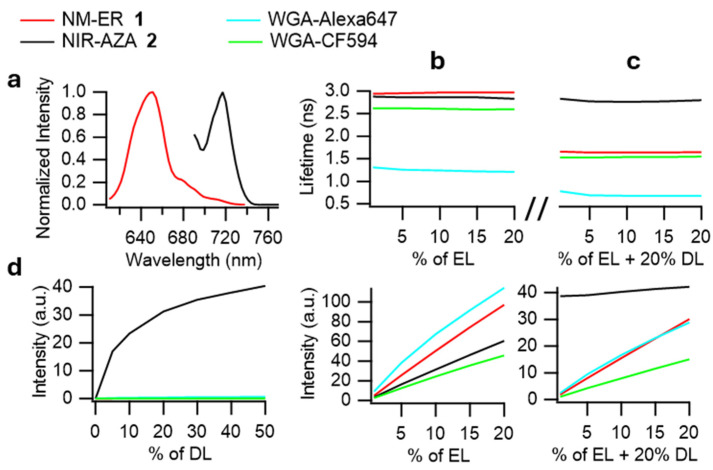
STED microscopy related photophysical properties of the fluorophores in solution. (**a**) Emission spectra of NM-ER **1** (excitation 594 nm) and NIR-AZA **2** (excitation 680 nm). Phasor lifetimes (top graph) and number of photons (bottom graph) of the different fluorophores measured upon excitation of the EL (**b**) without (% of EL) and (**c**) with 20% power of the DL (% of EL + 20% DL). (**d**) Number of photons collected upon “cross-excitation” with different power of the DL (% of DL).

**Figure 4 membranes-15-00009-f004:**
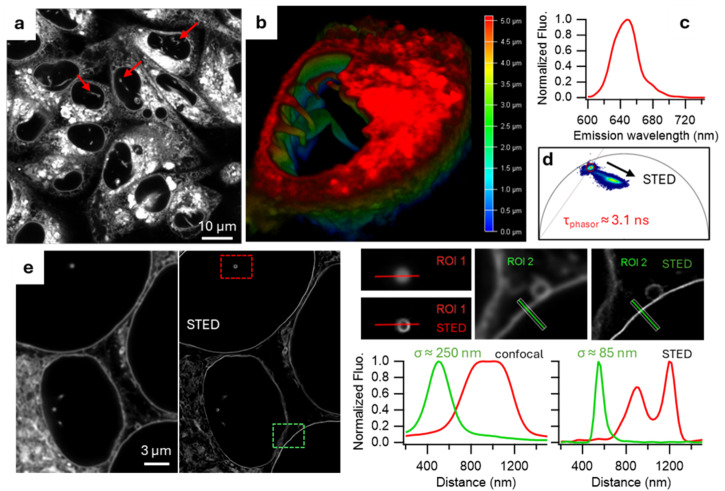
Characterization of the NM-ER fluorophore in cells. NM-ER fluorophores were incubated for at least 30 min in MDA (**a**–**d**) or HeLa cells (**e**). (**a**) Multiple MDA cells stained with NM-ER (red arrows indicationg invaginations) and (**b**) a 3D image of a nucleus (depth is color coded with 0 corresponding to the bottom of the cell near the coverslip, in blue). (**c**) Emission spectra measured in MDA cells. (**d**) Overlay of two FLIM phasor plots, one under 594 mm excitation with a cloud of pixels circled in red (τ ≈ 3.1 ns) and the other one with the depletion laser ON for STED imaging (cloud of dots spreading along the arrow). (**e**) Confocal (left) and STED (right) images comparison with 2 ROIs used for line plotted profile of intensities (ROI1 in red and ROI2 in green) with σ the FWHM.

**Figure 5 membranes-15-00009-f005:**
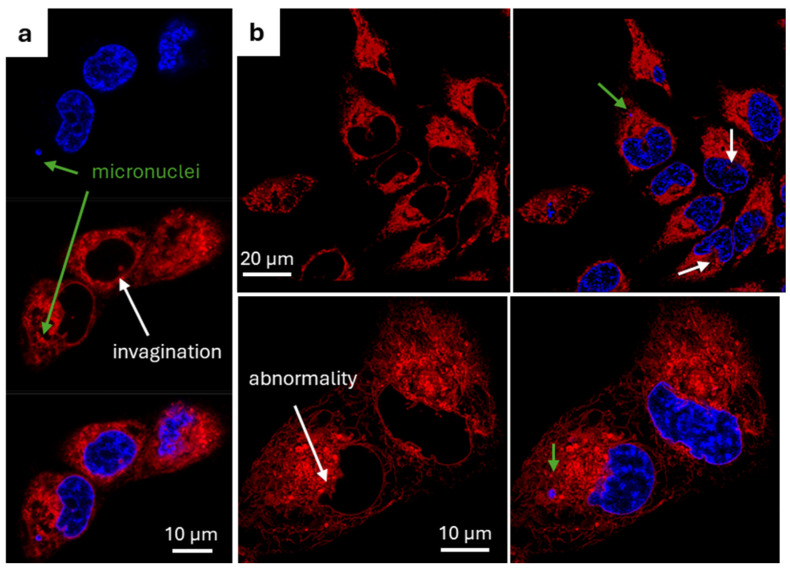
Different features observable with NM-ER. Confocal images of live cells co-stained with NM-ER fluorophore (red) and Hoechst (blue) showing some abnormalities in the shape of the nucleus envelope, nuclear invaginations (white arrows) and micronuclei (green arrows). (**a**) One field of view of MDA cells with separate detection channels and overlay. (**b**) Two different FOVs of HeLa (top images) and MDA cells (bottom images) with only NM-ER detection and the overlay with Hoechst.

**Figure 6 membranes-15-00009-f006:**
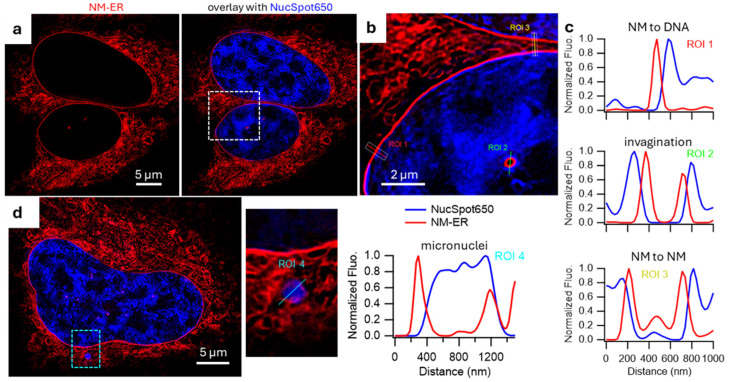
NM-ER and NucSpot650 STED images. Fixed HeLa cells stained with NucSpot650 (DNA, blue) and NM-ER fluorophore (red). (**a**) Detection channel for NM-ER (left panel) and overlay with detection channel of NucSpot650 (right panel). (**b**) Zoom of the overlay image with 3 ROIs plotted in (**c**). (**d**) Overlay showing a different cell with micronuclei with a zoom in of it and a ROI plot.

**Figure 7 membranes-15-00009-f007:**
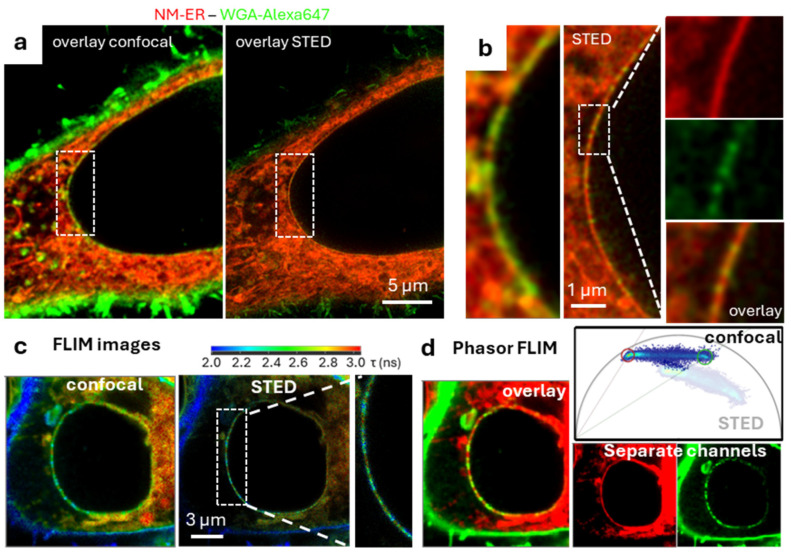
Fixed HeLa cells, co stained with NM-ER **1** and WGA-Alexa647 fluorophores. (**a**) Overlay of both detection channels (excitation at 594 nm of **1** in red and 680 nm for Alexa647 in green) in confocal (left image) and STED (right image). (**b**) Expanded highlighted inset sections of (**a**) with an expansion of a smaller section of the nucleus envelope with the two separate channels and the overlay in STED. (**c**,**d**) FLIM analysis of a cell using a single excitation wavelength at 594 nm and one detection channel. (**c**) FLIM image in confocal (left) and STED (right) with each pixel colored according to the average arrival time of photons (shorter lifetimes in blue and longer lifetimes in red). (**d**) Phasor analysis of the same cell. The red and green circle on the phasor plot correspond to the single phase lifetime of **1** and Alexa647, respectively and were used to separate the 2 channels (red and green images with the overlay).

**Figure 8 membranes-15-00009-f008:**
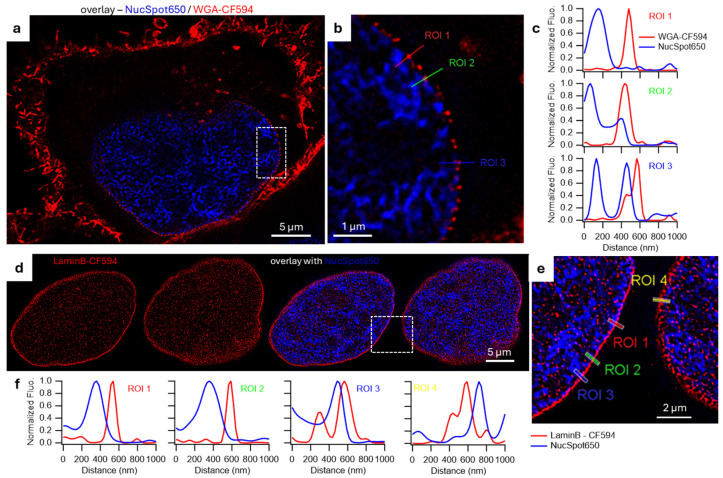
Comparison with other structures of the nucleus in super resolution. STED images of fixed HeLa cells (**a**,**b**) stained with NucSpot650 and WGA-CF594 or (**d**,**e**) permeabilized and stained with antibodies against LaminB and with NucSpot650. (**b**,**e**) zoom in with (**c**,**f**) intensities profile of the lines corresponding to the different ROIs.

**Table 1 membranes-15-00009-t001:** Microscope parameters used for cell image acquisition.

Fluorophore (i)	Exc 1 (nm)	Em 1 (nm)	Fluorophore (ii)	Exc 2 (nm)	Em 2 (nm)
NM-ER	594	610–740			
WGA-Alexa647	647	660–750			
WGA-CF594	594	605–690			
LaminB-CF594	594	605–690			
Hoechst	405	415–520			
NM-ER	594	605–655	WGA-Alexa647	680	690–750
NM-ER	594	605–655	NucSpot650	680	690–750
WGA-CF594	594	605–645	NucSpot650	650	665–700
LaminB-CF594	594	605–660	NucSpot650	665	685–750

**Table 2 membranes-15-00009-t002:** Microscope parameters used for in-solution measurements of fluorophores.

Fluorophore	Excitation (nm)	Detection Range (ns)	Phase Lifetime (ns)
NM-ER **1**	594	610–740	3
NIR-AZA **2**	685	700–750	2.8
WGA-Alexa647	647	660–750	1.2
WGA-CF594	594	605–690	2.6

## Data Availability

Data are contained within the article and Appendix A.

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
