# Peer review of "A STEDable BF2-Azadipyrromethene Fluorophore for Nuclear Membrane and Associated Endoplasmic Reticulum Imaging"

_membranes, 2025, doi:10.3390/membranes15010009_

Round 1

Reviewer 1 Report

Comments and Suggestions for Authors

This study presents a new molecular probe (NM-ER) for imaging the interconnection between the endoplasmic reticulum and the nuclear membrane, claiming to significantly improve resolution, especially compared to conventional confocal laser scanning microscopy. Overall, the research direction is of certain significance, but the innovation and experimental support still need further enhancement. Therefore, my recommendation is to request a major revision of the manuscript.

1.     In the introduction, the authors emphasize the design of a probe that can stain both the nuclear membrane and the endoplasmic reticulum. However, in the subsequent experiments, while the probe is capable of marking both the nuclear membrane and the endoplasmic reticulum, there is no further discussion on the significance of simultaneously labeling both compartments. In the later experiments on the recognition and quantification of nuclear abnormalities, the probe is primarily used for nuclear membrane localization, and the simultaneous localization of the endoplasmic reticulum seems somewhat redundant.

2.     In Figure 4, there is no co-localization experiment with a commercially available endoplasmic reticulum targeting probe, which makes it unclear how NM-ER's targeting specificity to the endoplasmic reticulum was determined. This raises some concerns regarding the claim. Additionally, besides demonstrating that the probe effectively localizes to the intended organelle, it is also necessary to provide evidence that it does not target other organelles, to ensure specificity.

3.     The authors provide only a 3D reconstruction experiment as evidence of the probe's photostability, but more data are needed. Specifically, the authors should provide cell imaging at different time points under prolonged exposure to the same laser to further validate the photostability of the probe under conditions that mimic actual usage. Additionally, details regarding the laser power used in these experiments should be included.

4.     Please provide data on the cytotoxicity of the probe and its interference with ions to demonstrate the probe's safety and specificity in biological systems.

5.     Please provide the NMR and MS spectra of the probe and include them in the manuscript to support the structural confirmation of the probe.

Author Response

Comment 1.  In the introduction, the authors emphasize the design of a probe that can stain both the nuclear membrane and the endoplasmic reticulum. However, in the subsequent experiments, while the probe is capable of marking both the nuclear membrane and the endoplasmic reticulum, there is no further discussion on the significance of simultaneously labeling both compartments. In the later experiments on the recognition and quantification of nuclear abnormalities, the probe is primarily used for nuclear membrane localization, and the simultaneous localization of the endoplasmic reticulum seems somewhat redundant. 

Response 1:  Thanks for the positive and insightful comments.  In the introduction we have explained that the endoplasmic reticulum is continuous with the nuclear membrane and therefore, NM-ER would be expected to stain both the nucleus membrane and the ER.  While our focus for this work was more the nuclear membrane, we felt it important to highlight the alternative potential use as an ER stain.  An additional reference has been added to the text to emphasise this as reference [1].

Comment 2.  In Figure 4, there is no co-localization experiment with a commercially available endoplasmic reticulum targeting probe, which makes it unclear how NM-ER's targeting specificity to the endoplasmic reticulum was determined. This raises some concerns regarding the claim. Additionally, besides demonstrating that the probe effectively localizes to the intended organelle, it is also necessary to provide evidence that it does not target other organelles, to ensure specificity. 

Response 2:  Thanks for this very helpful observation. We have carried out additional experiments and added the results as a supplementary figure showing co-staining of the ER and nucleus membrane with NM-ER and the green emitting ER-Tracker Green (Invitrogen, E34251) demonstrating a high colocalization with Pearson’s and Mander’s coefficients (Figure S1). 

Comment 3.  The authors provide only a 3D reconstruction experiment as evidence of the probe's photostability, but more data are needed. Specifically, the authors should provide cell imaging at different time points under prolonged exposure to the same laser to further validate the photostability of the probe under conditions that mimic actual usage. Additionally, details regarding the laser power used in these experiments should be included. 

Response 3:  Thanks for pointing out this omission in our submission. We have conducted photobleaching experiments in live and fixed cells using two different laser powers (lower and higher used for this work) and compared NM-ER fluorophore with the ER-Tracker Green with the data shown in Figure S2. This data shows excellent photostability for NM-ER. Moreover, a series of fifty STED images have been acquired to show it is robustness even under the high-power illumination of the depletion laser (Figure S2).  

Comment 4.  Please provide data on the cytotoxicity of the probe and its interference with ions to demonstrate the probe's safety and specificity in biological systems. 

Response 4:  Thanks for pointing this out to us.  Additional experiments have been carried out and included in the revised manuscript supporting information as SI Movie S2.  This shows an eight-hour timelapse of NM-ER stained HeLa cells, with images acquired every seven minutes demonstrating viable cells undergoing normal cell division throughout with no early death.  

Comment 5.  Please provide the NMR and MS spectra of the probe and include them in the manuscript to support the structural confirmation of the probe. 

Response 5:  Analytical spectrum included in the supporting information as Figure S8.

Reviewer 2 Report

Comments and Suggestions for Authors

Bourgès et al. report on super-resolution (STED) microscopy imaging of the nuclear membrane including nuclear pores as  well as the adjacent endoplasmic reticulum. For this purpose the authors developed a red emitting staining fluorophore with the advantages of low phototoxicity, little photobleaching and the principal possibility of high penetration depth within tissue. Experiments are combined with staining of DNA or antibodies by commercial fluorophores. Images of live cells as well as of fixed cells are convincing. However, the authors should specify the irradiance and exposure times for live cell imaging, since at typical light exposures for STED microscopy living cells are often severely damaged. The authors should comment on this issue in their discussion, also in the context that FLIM experiments need longer exposure times than conventional microscopy.

Minor issues:

- How appropriate is the depletion wavelength for STED at 775 nm, which is by more than 100 nm shifted towards the fluorescence maximum?

Abstract: Correct the phrase “… compared to comparable confocal laser scanning microscopy”.

Lines 117-118: The authors should define “fast FLIM”. How could the “average lifetime” deduced from these experiments be related to the so-called “phasor lifetime”? Perhaps the phasor technique would also deserve a few words of explanation for non-specialists.

Table 2: Specify the bandwidths for excitation. Since in the case of co-staining with several dyes the emission spectra were overlapping, how could the authors distinguish between the different dyes? Did they switch the excitation wavelength?

Fig. 3c: What does it mean, if the fluorescence intensity remains constant with increasing power of the excitation laser? Is it due to saturation?

Author Response

Comment 1:  Bourgès et al. report on super-resolution (STED) microscopy imaging of the nuclear membrane including nuclear pores as  well as the adjacent endoplasmic reticulum. For this purpose the authors developed a red emitting staining fluorophore with the advantages of low phototoxicity, little photobleaching and the principal possibility of high penetration depth within tissue. Experiments are combined with staining of DNA or antibodies by commercial fluorophores. Images of live cells as well as of fixed cells are convincing. However, the authors should specify the irradiance and exposure times for live cell imaging, since at typical light exposures for STED microscopy living cells are often severely damaged. The authors should comment on this issue in their discussion, also in the context that FLIM experiments need longer exposure times than conventional microscopy. 

Response 1:  Thanks for these very helpful observations. The materials and methods section has been expanded (see lines 103 to108) to now include exposure times and laser powers used for both whilte light and depletion lasers with additional citation included. Additional experiments have been carried out with data in the revised Supporting information, that shows that NM-ER is very photostable with minimal photobleaching occurring under STED imaging conditions. (Figure S2) Under the illumination used in this work, photobleaching is minimized. Photobleaching experiments and long timelapse have been measured (Figure S2-3 and Movie S2). 

Comment 2:  How appropriate is the depletion wavelength for STED at 775 nm, which is by more than 100 nm shifted towards the fluorescence maximum? 

Response 2:  The 775 nm depletion wavelength is ideal for NM-ER. Typically, excitable dyes with emission maxima from 625 to 700 nm are depleted with the 775 nm depletion laser. In our test experiments shown in Figure 3b and 3c we have shown that NM-ER is effectively depleted with this laser as are commercial fluorophores with similar wavelengths.  

Comment 3:   Abstract: Correct the phrase “… compared to comparable confocal laser scanning microscopy”.   Lines 117-118: The authors should define “fast FLIM”. How could the “average lifetime” deduced from these experiments be related to the so-called “phasor lifetime”? Perhaps the phasor technique would also deserve a few words of explanation for non-specialists. 

Response 3:  Thanks for this very helpful comment and agree it could cause confusion.  As such we have removed the term “fast” FLIM and replaced it with just FLIM.  Essentially, fast FLIM is the same as FLIM (from fitting the lifetime decay but it is a term only used in the Leica microscope software (the instrument used in this work)) so it is not appropriate. Explanations on the difference between FLIM and FLIM phasor have been added to the main text at lines 122-128 and 225. 

Comment 4:  Table 2: Specify the bandwidths for excitation. Since in the case of co-staining with several dyes the emission spectra were overlapping, how could the authors distinguish between the different dyes? Did they switch the excitation wavelength? 

Response 4:  Thanks for the opportunity to clarify these points. The bandwidths for excitation don’t apply here as lasers have been used.  In figure 7a and 7b, two excitations were used. To avoid cross excitation of NM-ER with the excitation of Alexa647 at 647 nm, its excitation was shifted to longer wavelength (680 nm, same approach with NucSpot650 and NM-ER). Alexa647 being also excited at 594 nm (excitation of NM-ER), only the photons at shorter wavelength were collected to prevent overlapping with the emissions of Alexa647. In figure 7c and 7d, both were excited at 594 nm and all photons were collected. We made use of their distinct lifetimes to separate them.

Comment 5:  Fig. 3c: What does it mean, if the fluorescence intensity remains constant with increasing power of the excitation laser? Is it due to saturation? 

Response 5:  Thanks for bringing this to our attention.  In answer, there was no saturation of the detector and the plateauing of the intensity we believe is an effect of the combination of anti-Stokes emission phenomena and depletion occurring. 

Round 2

Reviewer 1 Report

Comments and Suggestions for Authors

After carefully reviewing the manuscript, I find that the work is well-organized and presents the research clearly. I do not have any further comments or suggestions for improvement at this stage.